# Leveraging Technology to Diagnose Alzheimer’s Disease: A Systematic Review and Meta-Analysis

**DOI:** 10.3390/healthcare11233013

**Published:** 2023-11-21

**Authors:** Clemens Scott Kruse, Michael E. Mileski, Reagan Wilkinson, Britney Hock, Roger Samson, Taylor Castillo

**Affiliations:** School of Health Administration, Texas State University, San Marcos, TX 78666, USA

**Keywords:** health information technology, Alzheimer’s disease, dementia

## Abstract

Background: About 50 million people worldwide suffered from dementia in 2018—two-thirds of those with Alzheimer’s disease (AD). By 2050, this number is expected to rise to 152 million—which is slightly larger than the country of Russia. About 90% of these people are over the age of 65, but early-onset dementia can occur at younger ages. Early detection is imperative to expedient treatment, which can improve outcomes over the span of diagnosis. Objectives: To conduct a meta-analysis of similar studies along with a systematic literature review to hasten the development of clinical practice guidelines to assist clinicians in the diagnosis of AD. We analyzed data points in each article published over the last 10 years to meet this objective: cost, efficiency, accuracy, acceptability (by physician and patient), patient satisfaction, and barriers to adoption. Methods: Four research databases were queried (PubMed, CINAHL Ultimate, Web of Science, and ScienceDirect). The review was conducted in accordance with a published protocol, the Kruse Protocol, and reported in accordance with PRISMA (2020). Results: Ten interventions were identified to help diagnose AD among older patients, and some involved a combination of methods (such as MRI and PET). The average sample size was 320.32 (SD = 437.51). These 10 interventions were identified as accurate, non-invasive, non-stressful, inexpensive, convenient, and rapid. Only one intervention was identified as statistically ineffective, and this same intervention was used effectively in other studies. The barriers identified were cost, training, expense of travel, and required physical presence of patient. The weighted average sensitivity was 85.16%, specificity was 88.53, and the weighted average effect size was 0.7339 (medium). Conclusion: Innovation can accurately diagnose AD, but not all methods are successful. Providers must ensure they have the proper training and familiarity with these interventions to ensure accuracy in diagnosis. While the physical presence of the patient is often required, many interventions are non-invasive, non-stressful, and relatively inexpensive.

## 1. Introduction

### 1.1. Rationale

Prior to the pandemic, Alzheimer’s disease (AD) was growing into the largest fear for older adults. In the US, AD is fatal for more people than breast cancer and prostate cancer, and it is the number one cause of death in Great Britain. The 2018 prevalence of dementia was 50 million, but it is expected to surpass 150 million by 2050. Alzheimer’s disease accounts for about two-thirds of the dementia population [1].

The disease has no cure: there are only about 10 drugs approved to manage AD’s symptoms. The disease manifests itself in the form of plaque on the brain that affects the communication of about 100 billion neurons, which degrades and ultimately destroys these neurons. At first, this destruction manifests itself in simple forgetfulness of recently learned things, but over time, the destruction becomes more severe, affecting speech, motor movement, and long-term memory [2]. Medicine today does not fully understand the etiology and pathogenesis of AD [3]. Medicine can treat but cannot prevent or cure the disease [4,5]. For these reasons, many healthcare professionals have promoted research related to the detection of this disease because more timely diagnoses could potentially reduce the overall burden faced by society [2].

For a short period, diagnosis could only be confirmed through autopsy; however, health information technology (HIT) has made strides toward diagnosis while the patient lives. Optical and electrochemical biosensors are being developed to sense biological responses and identify biomarkers for AD [6]. Canada recently published a white paper on the early diagnosis of AD. It included questionnaires and cognitive testing, diagnosis of neurodegeneration through MRI, and established biomarkers for neurodegeneration and Alzheimer’s pathology through imaging and biomarkers [7]. Wearables and other remote sensors track behavior and can help detect neurodegenerative diseases like AD in a timely and economical manner [8].

A systematic review was published in 2022 that evaluated apps, sensors, and virtual reality developed to help with timely diagnosis, management, and treatment of symptoms. Eight studies were analyzed from the last decade to identify solutions: deficits in finger dexterity, memory retrieval, and alertness and mood improvement [9]. This study did not identify how leveraging health information technology can diagnose AD.

A systematic review was published in 2021 that analyzed health monitoring and artificial intelligence (AI) for deep learning. This review found AI can be used in the early detection of chronic diseases. Cloud computing was a catalyst for this innovation, and the integration of a blockchain framework improved data security to help prevent the misuse of patient data [10]. This study did not identify how leveraging health information technology can diagnose AD.

### 1.2. Objectives

The purpose of this review is to conduct a meta-analysis of similar studies along with a systematic literature review to hasten the development of clinical practice guidelines to assist in the diagnosis of AD. To meet this objective, we analyzed several data points in each article published over the last 10 years. The intention was to analyze studies with strong methodological approaches to identify trends of effectiveness. This approach should give providers and families of those suffering from AD a pathway to a non-invasive, cost-effective technique to confirm an AD diagnosis. Findings from this review will hasten practice guidelines for rapid, inexpensive, and accurate diagnosis of AD.

## 2. Methods

### 2.1. Eligibility Criteria

To be eligible for consideration for this systematic literature review, studies had to use as participants older adults who were undergoing a diagnosis for AD, in a study published in the last ten years, published in a peer-reviewed journal, using strong methodologies. A randomized controlled trial (RCT) was preferred, but we also allowed quasi-experimental, mixed methods, quantitative, and qualitative. A wide variety of interventions were preferred to calculate an overall effect size. This included physical markers, digital markers, and telemedicine interventions. Artificial intelligence interventions such as Bayesian networks and machine learning were also accepted. Machine learning often uses test sets to train the computer in what a positive diagnosis looks like, such as structural equation modeling (SEM). Then, the training is used on a larger set of data to identify the disease. Other systematic reviews were not included in the analysis so as not to confuse the results (because systematic reviews already reported on results from studies that may also be counted in our analysis). Studies must use humans as subjects, be published in the English language, and have a full-text version available for download to enable data extraction. We did not allow gray literature, editorials, protocols, or any other article that did not report study results. This review is registered with PROSPERO: ID 350266. The protocol used for this meta-analysis can be found at Kruse CS. Writing a systematic review for publication in a health-related degree program. JMIR research protocols. 2019 Oct 14;8(10):e15490 [11].

### 2.2. Information Sources

We queried four research databases PubMed (MEDLINE), CINAHL Ultimate (excluding MEDLINE), Web of Science (excluding MEDLINE), and ScienceDirect (excluding MEDLINE). These databases were searched between 25 July 2022 and 26 July 2022. MEDLINE was excluded from all databases except PubMed to help eliminate duplicates.

### 2.3. Search Strategy

We used the Medical Subject Headings (MeSH) of the U.S. Library of Medicine to create a Boolean search string combining key terms listed in the literature: “Alzheimer Disease” AND diagnosis AND (technology OR “artificial intelligence” OR mhealth). We used the same search strategy in all databases and employed the same filters, where available.

### 2.4. Selection Process

We searched for key terms in all databases, filtered results, and screened abstracts for applicability in accordance with the Kruse Protocol [11]. At least two reviewers screened all abstracts and rejected any study that did not report results (e.g., protocols, editorials, etc.). Three consensus meetings were held to determine which articles would be analyzed, which data-extraction items were significant measurements of effectiveness, and what observations should become themes. A kappa statistic was calculated. Results were reported in accordance with the PRISMA 2020 standard [12].

### 2.5. Data Collection Process

We used a standardized Excel spreadsheet as a data extraction tool, collecting key data items in each article. This spreadsheet was standardized in the published protocol, and it provides fields of value for administrators, clinicians, and policy makers [11].

### 2.6. Data Items

In accordance with the published protocol, we collected the following fields of data at each step: PICOS (Participant demographics, Intervention specifics, Comparison of results between intervention and control, specific medical Outcomes, and Study design), observed bias, effect size, sensitivity, specificity, F1, country of origin, statistics used, strength of evidence, quality of evidence, measures of effectiveness (cost, efficiency, invasiveness, etc.), and barriers to adoption. A narrative analysis was conducted to identify themes in the literature [13].

### 2.7. Study Risk of Bias Assessment

As each article was analyzed, reviewers noted items of bias (selection bias, sample bias, etc.). We assessed the quality of each study using the John’s Hopkins Nursing Evidence Based Practice tool (JHNEBP) [14]. Instances of bias helped interpret the results because bias can limit external validity.

### 2.8. Effect Measures

Because reviewers accepted mixed methods and qualitative studies, they were unable to standardize summary measures, as would be performed in a meta-analysis. Measures of effect were summarized in tables where they were reported. Odds ratios, correlation coefficients, and the F1 were converted to Cohen’s d [15,16]. A weighted average effect size, group specificity, and sensitivity were reported.

### 2.9. Synthesis Methods

Sensitivity, specificity, F1, and effect size were collected from studies. A meta-analysis of this data was performed. Reviewers also performed a thematic analysis of the data collected from studies. Same or similar observations were combined into themes. Themes and other observations were tabulated into affinity matrices.

### 2.10. Reporting Bias Assessment

The overall ratings of quality and strength of evidence, identified using the JHNEBP, provided us with an assessment of the applicability of the cumulative evidence. Reviewers also noted instances of bias such as selection bias, sample bias, and publication bias.

### 2.11. Additional Analyses and Certainty Assessment

A narrative analysis converted observations into themes (common threads between articles) [13]. We calculated the frequency of occurrence and reported this in a series of affinity matrices. The frequency reporting provides confidence in the analyzed data.

## 3. Results

### 3.1. Study Selection

Figure 1 provides the study selection process to include the inclusion and exclusion criteria from the four research databases. A kappa statistic was calculated based on the levels of agreement between reviewers (*k* = 0.83, strong agreement). Twenty-eight studies were selected for the systematic literature review. Based on the data provided in these articles, only 15 qualified for the meta-analysis.

### 3.2. Study Characteristics

Following the PRISMA (2020) checklist, a PICOS table was created to summarize the study characteristics of the 28 studies. This is tabulated in Table 1. Of the 28 studies analyzed over the 10-year period, zero were from 2012, three were from 2013 [17,18,19], two were from 2014 [20,21], zero were from 2015, one was from 2016 [22], two were from 2017 [23,24], five were from 2018 [25,26,27,28,29], three were from 2019 [30,31,32], three were from 2020 [33,34,35], seven were from 2021 [36,37,38,39,40,41,42], and two were from 2022 [43,44]. All involved older adults, 61% were quasi-experimental, 21% were observational, and 14% were true experiments. Of those analyzed, 9/28 (32%) used positron emission tomography (PET) to examine tau (plaque) on the brain, 6/28 (21%) examined blood-based biomarkers, 4/28 (14%) used some form of telehealth (video teleconferencing, virtual reality, or telemonitoring), 3/28 (11%) used artificial intelligence or machine learning to recognize patters from MRI or PET scans, while the other six interventions stood on their own without repeats (MRI, MRI + PET, arterial pulse, RNA, EEG, and spectral domain optical coherence tomography). The average sample size was 320.32 (SD = 437.51).

### 3.3. Risk of Bias within and across Studies

We used the JHNEBP quality assessment tool to identify the strength and quality of the evidence. Of the group of articles analyzed, 18/28 (64%) were strength II (quasi-experimental), 6/28 (21%) were strength III (non-experimental, qualitative, or meta-analysis), and 4/28 (14%) were strength I (experimental study, RCT). Additionally, 26/28 (93%) were quality A (consistent results with sufficient sample sizes, adequate control, definitive conclusions), and 2/28 (7%) were quality B (well-defined, reproducible search strategies, consistent results with sufficient numbers of well-defined studies, and definitive conclusions). Most articles analyzed were high-quality with strong methods.

Reviewers also noted instances of bias. The most common bias identified was selection bias. Every study (100%, 28/28) analyzed in this review was conducted in one region of one country, creating a selection bias. Additionally, in 6/28 (21%) of the studies analyzed, instances of sample bias were identified because the sample was heavily skewed with one gender or race.

### 3.4. Results of Individual Studies

Following the Kruse Protocol, reviewers independently recorded observations from each article, commensurate with the objective. Once all reviewers had recorded their observations, a consensus meeting was held to discuss the findings. The result of this meeting was a thematic analysis: making sense of the data. When reviewers noted an observation multiple times, it was recorded as a theme to summarize the observations. These themes are tabulated in Table 2 along with the strength and quality assessments. A match of observation to theme is provided in Appendix A. A summary of other observations can be found in Appendix B.

### 3.5. Results of Syntheses

In 13 articles, effect size was not reported, which eliminated them from the meta-analysis. In 15/28 studies (54%), sufficient data were provided to conduct a meta-analysis. The necessary fields were specificity, sensitivity, F1, odds ratio, Pearson’s *r*, and Cohen’s *d* for effect size. Ten articles reported sensitivity and specificity [17,19,20,26,28,30,33,38,39,42]. The overall sensitivity was 0.8516, specificity was 0.8853, precision (PPV) was 0.8388, accuracy was 0.9074, F1 = 0.8452, OR = 44.31 CI [35.53, 55.28]. The confusion matrix is illustrated in Figure 2. True positives, false negatives, true negatives, and false positives were collected, and odds ratios were calculated. All were converted to Cohen’s *d* [15,16]. The weighted average effect size was $\bar{d}$ = 0.7339, which is a medium effect size that approaches a large effect size. This means the weighted average effect size has a strong medium effect: health information technology interventions can diagnose AD using non-invasive methods. The full table and calculations are listed in Appendix C.

### 3.6. Additional Analysis and Certainty of Evidence

A series of affinity matrices were created to summarize the additional analysis. Themes and observations were organized this way to reflect the probability of their occurrence in the group for analysis.

### 3.7. Interventions of HIT to Diagnose AD

Table 3 summarizes the interventions observed. Four themes and six individual observations were identified by the reviewers for a total of 28 occurrences in the literature. PET was identified in 9/28 (32%) of the studies analyzed [23,27,29,32,36,37,40,41,43]. PET biomarkers can identify tau plaque on the brain. This intervention is less invasive, but the confined space of the machine can cause stress. Blood-based biomarkers were identified in 6/28 (21%) of the studies analyzed [17,25,31,34,39,42]. This intervention is more invasive than imaging and causes more risk. Various forms of telehealth were implemented to diagnose AD in 4/28 (14%) of the studies analyzed [20,24,33,44]. These ranged from video teleconference (VTC) and virtual reality to other televisits to assess cognition. It is important to note that results through telemedicine were the same as traditional visits, which means a clinic can expand productivity without expanding the physical plant, and it exposes the patient to less risk or iatrogenic illness or injury. Multiple forms of artificial intelligence (Bayesian networking, machine learning) were utilized in 3/28 (11%) of the articles analyzed [19,22,28]. These interventions utilized PET biomarkers, MRI biomarkers, or arterial spin diagnostics. Six other interventions occurred only once in the literature (21%). These included MRI biomarkers only, MRI and PET biomarkers combined, arterial pulse, RNA, EEG, and spectral domain optical coherence tomography [18,21,26,30,35,38].

Table 4 summarizes the effectiveness indicators observed. Five themes and three individual observations were identified by the reviewers for a total of 53 occurrences in the literature. The most identified theme was that the interventions were accurate in detecting/diagnosing AD. This occurred in 21/53 (40%) occurrences [17,18,19,20,21,22,23,24,26,28,32,34,35,36,37,38,39,40,42,43,44]. In 11/53 (21%) occurrences, the intervention was identified as non-invasive [20,21,25,26,27,28,33,35,36,38,41]. In 8/53 (15%) of the occurrences, two themes emerged: the intervention was identified as non-stressful for the patient [25,26,27,28,33,35,36,41], and the intervention was identified as inexpensive to implement [20,24,26,28,30,31,38,44]. In 2/53 (4%) of the occurrences, the intervention could be repeated without degradation of the results [26,41]. Three observations only occurred once in the literature: convenient, rapid, and ineffective [29,31,38]. Interestingly, the same intervention (PET biomarkers) was used successfully in other studies.

Table 5 summarizes the barriers observed. Five themes and two individual observations were recorded. The most common theme was cost. This occurred in 23/91 (25%) of the occurrences [17,18,19,20,21,22,23,27,28,30,31,32,33,34,35,36,37,39,40,42,43,44]. The need for training was identified in 21/91 (23%) of the occurrences [17,18,19,20,21,22,23,24,26,27,28,30,31,32,33,34,37,38,39,40,42]. In 18/91 (20%) of the occurrences, two themes were identified: requires the physical presence of the patient, and travel expenses are incurred (correlation of these two themes is high) [17,18,21,23,27,28,30,31,32,34,35,36,37,39,40,41,42,43]. In 9/91 (10%) of the occurrences that involved a blood draw, it was identified that taking blood is both invasive and incurs risk to both the patient and the provider [17,22,30,31,34,37,39,40,42]. One intervention involved post-mortem examinations to assess neurodegeneration of brain cells, so it was noted that a procedure to diagnose the living needs to be developed [29].

## 4. Discussion

A meta-analysis of similar studies was conducted along with a systematic literature review to identify ways to hasten the development of clinical practice guidelines to assist in the diagnosis of AD. Of the 28 studies analyzed, 10 interventions were identified. The interventions were effective at detection of AD in all but one study [29]. PET biomarkers [23,27,29,32,36,37,40,41,43] and blood-based biomarkers [17,25,31,34,39,42] accounted for more than 50% of the interventions. PET biomarkers were also used in combination with MRI biomarkers in one study [35]. While interventions were found to be effective at detecting AD, accuracy was only noted in 40% of the interventions [17,18,19,20,21,22,23,24,26,28,32,34,35,36,37,38,39,40,42,43,44]. Interventions were found to be non-invasive, non-stressful, relatively inexpensive, repeatable without degradation of results, convenient, and rapid. While most interventions required the physical presence of the patient, there were multiple interventions, such as AI and VTC that did not [19,20,22,24,28,33,44]. AI analyzed existing data, and VTC allowed providers to detect AD with surprising accuracy.

Accurate and early diagnosis of AD is as important to the patient as it is to their family [45]. The patient needs to know about pending secondary effects such as cognitive decline, sleep disruption, or stroke incident to AD. Their families need to know of AD detection to prepare for supportive relationships and quality of life. Providers need to know about the effectiveness of the 10 interventions identified in this research and their relative level of effectiveness ($\bar{d}$ = 0.7339). The relative effectiveness of these interventions (PET biomarkers, blood-based biomarkers, telehealth, AI/machine learning, MRI biomarkers, PET biomarkers) should lead to new clinical practice guidelines (CPGs). Policy makers need to develop reimbursement mechanisms for the most effective early detection interventions. Many interventions were listed as relatively inexpensive, but medical procedures in general are beyond the out-of-pocket reach of most patients or their care givers. Researchers need to know about early detection interventions to identify more potential participants in the research of this disease to seek treatments and a cure. Many of the interventions are shown to be highly accurate, non-invasive, non-stressful to the patient, repeatable, convenient, and rapid. These considerations can also be utilized to effect change to CPGs in that they show justifiable reasons for the changes. Despite barriers to implementation such as cost, training, physical presence of the patient being necessary, and travel, the aforementioned interventions provide tools to assist in the accurate diagnosis of AD.

Future research needs to expand some of these interventions into RCTs to firmly establish new CPGs. These new CPGs need to be advertised as widely available to patients and their families. Caring for those with dementia often requires significant changes to lifestyle, habits, and spending. Families need notice to make these adjustments to prepare for their new role and new relationship with the family member.

### Limitations

A limitation of this review was the length of time we looked back in the literature (10 years). We chose ten years because it should have identified the most recent advancements in diagnosis intervention. However, a longer time frame may have identified additional interventions. Another limitation is publication bias. We only looked at published works. There are possibly more successful interventions that have not yet been published. Another limitation is that we eliminated editorials and opinion articles because these did not provide empirical evidence. It is possible additional interventions could have been identified from gray literature.

## 5. Conclusions

Ten interventions from the literature from the last 10 years were identified as rapid and effective to diagnose AD. While some of these interventions are riskier and involve a blood draw, others are non-invasive and non-stressful to the patient. Additional research needs to expand these interventions into robust RCTs to establish new CPGs for the medical community.

## Figures and Tables

**Figure 1 healthcare-11-03013-f001:**
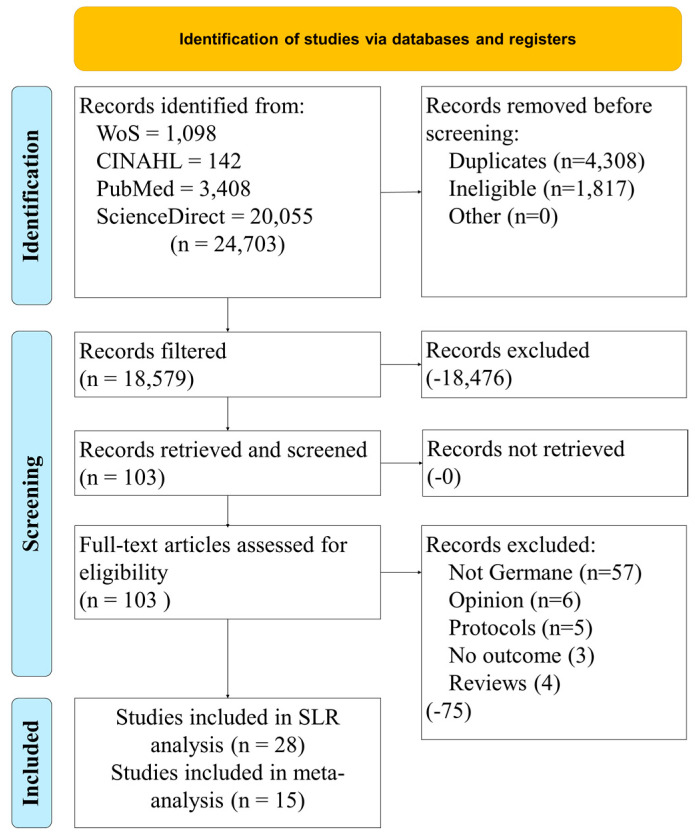
Study selection process.

**Figure 2 healthcare-11-03013-f002:**
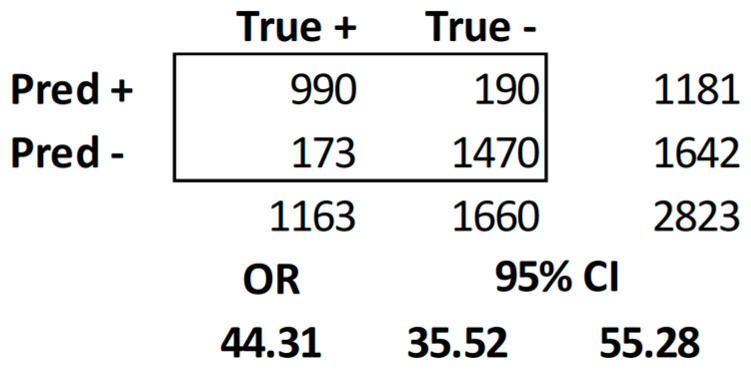
Confusion matrix.

**Table 1 healthcare-11-03013-t001:** PICOS.

Authors	Participants	Experimental Intervention	Results	Medical Outcomes Reported	Study Design
Guo et al. [17]	165 older adults with AD	Blood biomarkers (proteomics) in blood plasma to distinguish early AD from physiological aging and diagnose AD	A set of 5 plasma proteins was identified, which differentiated between the CON group and the AD dementia group	Positive predictability. A biological pathway analysis showed that 4 of 5 proteins belonged to a common network with amyloid precursor protein and tau	Quasi-experimental
Kirbas et al. [18]	80 older adults with AD, mean age 68.9	Spectral domain optical coherence tomography (SD-OCT) to measure retinal nerve fiber layer (RNFL) thickness to diagnose AD	The average RNFL thickness was significantly less in the AD patients than in controls (65 ± 6.2 mm vs. 75 ± 3.8 mm; *p* = 0.001)	Positive predictability. AD degrades the thickness of the retinal nerve. SD-OCT can be used to positively diagnose AD in early stages of disease	Quasi-experimental
Wang et al. [19]	181 older adult AD patients	Bayesian network (BN) analysis based on regional gray matter volumes to identify differences in structural interactions among core default mode network (DMN) regions in structural MRI data to diagnose AD	The structural interactions between the medial prefrontal cortex (mPFC) and other brain regions, including the left inferior parietal cortex (IPC), the left inferior temporal cortex (ITC), and the right hippocampus (HP), were significantly reduced in the AD group	The BN models significantly distinguished AD patients from normal controls	Quasi-experimental
Munro et al. [20]	202 older adults	Telepsychiatry and telepsychology through video teleconferencing (VTC) to diagnose AD	Highly similar results across VTC and in-person conditions	VTC is a valid method to diagnose AD	True experiment
Zou et al. [21]	40 older adults, average age 64.8, 58% female	MRI and MR spectroscopy (MRS) biomarkers to detect changes in arterial blood flow to diagnose AD	Significant difference in the mean MMSE scores between the AD group and the healthy control group	Accurate predictor of AD; CBF in the bilateral frontal region showed a significant decrease in the AD group	Quasi-experimental
Collij et al. [22]	260 older adults	Machine learning (AI) to arterial spin labeling to diagnose AD	Single-subject diagnosis in the prediction set by using the discrimination maps yielded excellent performance for AD versus SCD (AUC, 0.96; *p* < 0.01), good performance for AD versus MCI (AUC, 0.89; *p* < 0.01), and poor performance for MCI versus SCD (AUC, 0.63; *p* = 0.06). Application of the AD versus SCD discrimination map for prediction of MCI subgroups resulted in good performance for patients with MCI diagnosis converted to AD versus subjects with SCD (AUC, 0.84; *p* < 0.01) and fair performance for patients with MCI diagnosis converted to AD versus those with stable MCI (AUC, 0.71; *p* > 0.05)	With automated methods, age- and sex-adjusted ASL perfusion maps can be used to classify and predict diagnosis of AD, conversion of MCI to AD, stable MCI, and SCD with good to excellent accuracy and AUC values	Quasi-experimental
Hornberger et al. [23]	42 older adults	Positron emission tomography (PET) biomarkers to estimate AB neurotic plaque density to diagnose AD	Aβ-PET used as an adjunct to standard diagnostic assessment increased QALYs by 0.021 years and 10-year costs by EUR 470 per patient. The ICER was EUR 21,888 per QALY gained compared to standard diagnostic assessment alone. When compared with CSF, Aβ-PET costs EUR 24,084 per QALY gained	Aβ-PET was consistently cost-effective relative to the commonly used affordability threshold (EUR 40,000 per QALY).	
Zhou et al. [24]	30 older adults, average age 82.2, 43.3% female	Instrument trail-making task (iTMT) using a wearable sensor to identify motor-cognitive impairment and diagnose AD	Good-to-excellent reliability was achieved for all iTMT tests. Between-group difference was more pronounced when using iTMT. Pairwise comparison suggested strong effect sizes	Simple, safe, and practical iTMT system with promising results to identify cognitive and dual-task ability impairment among older adults	True experiment
Ashton et al. [25]	160 older adults	Saliva biomarkers, including tau, to diagnose AD	No median difference in salivary t-tau concentration was found between AD and mild cognitive impairment or healthy elderly control	Not a viable method to diagnose AD	Quasi-experimental
Babiloni et al. [26]	83 older adults, average at 74.7, 82% male	Resting state electroencephalographic (rsEEG) rhythms to diagnose AD	ADMCI and DLBMCI patients showed different features of cortical neural synchronization at delta and alpha frequencies underpinning brain arousal and vigilance in quiet wakefulness	Viable testing method with low level of invasiveness	Quasi-experimental
Jones et al. [27]	284 older adults, >50	PET biomarkers in microtubule-associated protein tau (MAPT) to diagnose AD	Tau PET signal was qualitatively and quantitatively different between participants with AD, clinically normal (CN) participants, and MAPT mutation carriers, with the greatest signal intensity in those with AD and minimal regional signal in MAPT mutation carriers with mutations in exon 10	Viable testing method with low level of invasiveness. Tau PET shows higher magnitude of binding in MAPT mutation carriers who harbor mutations that are more likely to produce AD-like tau pathology	Quasi-experimental
Lee et al. [28]	1342 older adults	Machine learning (AI) of MRI scans to assess cortical atrophy and diagnose AD	aMCI-converts had higher atrophy similarity at both baselines. Similarly, AD patients with faster decline had higher atrophy similarity than slower decliners at baseline	The AD-specific atrophy similarity measure is a novel approach for the prediction of dementia risk and for the evaluation of AD trajectories on an individual subject level	Quasi-experimental
Lowe et al. [29]	687 older adults > 50	Tau-PET biomarkers to understand neurofibrillary tangle development to diagnose AD	Age-related elevated tau signal was seen among those with normal or abnormal amyloid status as compared to younger cognitively unimpaired individuals. Tau-PET signal increases modestly with age throughout the brain in cognitively unimpaired individuals, and elevated tau is seen more often when amyloid brain accumulation is present	Distinct patterns of neurofibrillary tangle deposition in younger-onset Alzheimer’s disease dementia versus older-onset Alzheimer’s disease dementia provide evidence for variability in regional tangle deposition patterns and demonstrate that different disease phenotypes have different patterns of tauopathy	Quasi-experimental
Fotuhi et al. [30]	81 older adults	Using RNA as blood-based biomarker to diagnose AD	Significant differences between AD subgroups and control in the whole plasma samples	Plasma BACE1-AS level may serve as a potent blood-based biomarker for Alzheimer’s disease	Quasi-experimental
Pase et al. [31]	1453 older adults > 65, average age 75, 54.5% female	Plasma total tau as a blood biomarker to diagnose AD	Higher plasma total tau level was associated with poorer cognition across 7 cognitive tasks and smaller hippocampi as well as neurofibrillary tangles and microinfarcts. Plasma t-tau shows early promise as a predictive biomarker for dementia	The findings suggest that plasma total tau levels may improve the prediction of future dementia, are associated with dementia endophenotypes, and may be used as a biomarker for risk stratification in dementia prevention trials	Observational
Tahmi et al. [32]	52 older adults	PET biomarkers for quantifying amyloid-B plaques on the brain to diagnose AD	No control group. Consistently and significantly higher SUVRs in comparison to the conventional method in almost all regions of interest	Processing the amyloid-β PET data in subjects’ native space can improve the accuracy of the computed SUVRs	Observational
Cabinio et al. [33]	139 older adults > 65	Virtual reality to assess memory decline to diagnose AD	SASG outperformed the Montreal cognitive assessment test (MoCA) in the ability to detect neuronal degeneration in the hippocampus on the right side	SASG is an ecological task, which can be considered a digital biomarker providing objective and clinically meaningful data about the cognitive profile of aMCI subjects	Quasi-experimental
Rajan et al. [34]	1327 older adults, 60% female, 60% African American	Blood biomarkers total tau (t-tau), neurofilament light (Nf-L), and glial fibrillary acidic protein (GFAP) to diagnose AD	Higher concentrations of serum biomarkers were associated with the development of clinical AD. Serum biomarkers were associated with faster cognitive decline over 16 years. Additionally, higher baseline t-tau was associated with faster increase in 3rd ventricular volume, and baseline Nf-L and GFAP were associated with faster decline in cortical thickness	Serum t-tau, Nf-L, and GFAP predict the development of sporadic AD and cognitive decline and changes in structural brain characteristics, suggesting their usefulness not only as screening and predictive biomarkers but also in capturing the pathogenesis of Alzheimer’s dementia.	Observational
Thientunyakit et al. [35]	51 older adults	MRI and PET biomarkers used to assess amyloid levels, glucose metabolism, and morphologic change in brain to diagnose AD	A significant direct linear correlation was observed between the AV45/FDG/NVol index and ADAS-Cog test score and an inverse correlation with TMSE score at baseline and with the degree of changes in ADAS and TMSE scores assessed one year later (disease progression)	V45/FDG/NVol index mapping of the brain is a novel quantitative molecular imaging biomarker that correlates with clinical neurocognitive status and may facilitate more accurate diagnosis, staging, and prognosis of AD	Quasi-experimental
Altomare et al. [36]	136 older adults	Using amyloid-PET and tau-PET biomarkers to diagnose AD	Amyloid-PET and tau-PET, when presented as the first exam, resulted in a change in etiological diagnosis in 28% and 28% of cases, and diagnostic confidence increased by 18% and 19%, respectively. When added as the second exam, amyloid-PET and tau-PET resulted in a further change in etiological diagnosis in 6% and 9% of cases, and diagnostic confidence increased by 4% and 5%, respectively	Amyloid-PET and tau-PET significantly impacted diagnosis and diagnostic confidence in a similar way, although a negative amyloid-PET has a stronger impact on diagnosis than a negative tau-PET. Adding either of the two as second exam further improved diagnostic confidence	True experiment
Desai et al. [37]	1159 older adults, 63% female	Tau concentration biomarkers based on activity levels to treat AD	No control. Participants with high total tau concentrations with medium physical activity had a 58% slower rate of cognitive decline, and those with high physical activity had a 41% slower rate of cognitive decline. Among participants with low total tau concentrations, medium physical activity was associated with a 2% slower rate of cognitive decline, and high physical activity was associated with a 27% slower rate of cognitive decline, compared with little physical activity	Those with high physical activity had a slower rate of cognitive decline	Observational
Lin et al. [38]	161 older adults	Arterial pulse spectrum and multilayer-perception analysis to diagnose classification of AD	Radial blood pressure waveform (BPW) indices differed significantly between the AD patients (6247 pulses) and control subjects (6626 pulses). Significant intergroup differences were found between mild, moderate, and severe AD	BPW can help classify extent of AD	Quasi-experimental
Liu et al. [39]	1989 older adults	Serum miR-24-3P biomarkers to diagnose AD	Expression of miR-24-3p showed 1.6-fold increase in AD group compared with healthy controls, and a negative correlation of miR-24-3p with mini-mental state examination score was obtained	MiR-24-3p has a certain value in the diagnosis of AD and may be a potential biomarker	Quasi-experimental
Mila-Aloma et al. [40]	318 older cognitively unimpaired adults, average age 60.5	Using amyloid-B biomarker pathology to classify cognitively unimpaired individuals and diagnose AD	There are biologically meaningful Aβ-downstream effects in individuals with a low burden of Aβ pathology, while structural and functional changes are still subtle or absent	These findings support considering individuals with a low burden of Aβ pathology for clinical trials	Observational
Sajjad et al. [41]	136 older adults	Using PET biomarkers and synthetic data augmentation (DCGAN model) to diagnose AD	With a 72% accuracy, the computer was able to identify normal, MCI, and AD images	AI can help diagnose AD from PET scans	True experiment
Wu et al. [42]	159 older adults, average age 68.5, 58% female	Plasma biomarker (p-tau and t-tau) to diagnose AD	p-tau181 had the greatest potential for identifying patients with cognitive impairment	Simplified diagnostic model provides an accessible and practical way for large-scale screening in the clinic and community, especially in developing countries	Quasi-experimental
Chun et al. [43]	25 older adults	18F-THK5351 PET biomarkers to diagnose AD	The patients in the 18F-THK5351-positive group were older than those in the 18F-THK5351-negative group	The results of the present study suggest that increased 18F-THK5351 uptake might be a useful predictor of poor prognosis among Aβ–aMCI patients, which might be associated with increased neuroinflammation	Observational
Kim et al. [44]	18 older adults	Telemonitoring to diagnose AD	It was effective in predicting dementia risk, with up to an 0.99 area under the curve (AUC) using DNN with principal component analysis (PCA) and a quantile transformer scaler	Can be used for a variety of long-term monitoring and early symptom detection systems, helping caregivers provide optimal interventions to elderly individuals at risk for dementia	Quasi-experimental

**Table 2 healthcare-11-03013-t002:** Summary of analysis sorted most recent to oldest.

Authors	Intervention Theme	Effectiveness Themes	Barrier Themes	Strength of Evidence	Quality of Evidence
Guo et al. [17]	Blood biomarkers	Accurate at detecting AD	Taking blood is invasive	II	A
			Cost of intervention		
			Must train users		
			Requires physical presence of patient		
			Expense of travel involved		
Kirbas et al. [18]	Spectral domain optical coherence tomography	Accurate at detecting AD	Cost of intervention	II	A
			Must train users		
			Requires physical presence of patient		
			Expense of travel involved		
Wang et al. [19]	AI/machine learning	Accurate at detecting AD	Cost of intervention	II	A
			Must train users		
Munro et al. [20]	Telehealth/VTC/VR	Non-invasive	Cost of intervention	I	A
		Accurate at detecting AD	Must train users		
		Inexpensive			
Zou et al. [21]	MRI biomarkers	Non-invasive	Cost of intervention	II	A
		Accurate at detecting AD	Must train users		
			Requires physical presence of patient		
			Expense of travel involved		
Collij et al. [22]	AI/machine learning	Accurate at detecting AD	Taking blood is invasive	II	A
			Cost of intervention		
			Must train users		
Hornberger et al. [23]	PET biomarkers	Accurate at detecting AD	Cost of intervention	II	A
			Requires physical presence of patient		
			Expense of travel involved		
			Must train users		
Zhou et al. [24]	Telehealth/VTC/VR	Accurate at detecting AD	Must train users	I	A
		Inexpensive			
Ashton et al. [25]	biomarkers	Non-invasive	Not reported	II	A
		Non-stressful			
Babiloni et al. [26]	EEG	Inexpensive	Must train users	II	A
		Non-invasive			
		Non-stressful			
		Repeatable without degradation of results			
		Accurate at detecting AD			
Jones et al. [27]	PET biomarkers	Non-invasive	Cost of intervention	II	A
		Non-stressful	Must train users		
			Requires physical presence of patient		
			Expense of travel involved		
Lee et al. [28]	AI/machine learning	Inexpensive	Cost of intervention	II	A
		Accurate at detecting AD	Must train users		
		Non-invasive	Requires physical presence of patient		
		Non-stressful	Expense of travel involved		
Lowe et al. [29]	PET biomarkers	Ineffective	Need a procedure for the living	II	A
Fotuhi et al. [30]	RNA	Inexpensive	Taking blood is invasive	II	A
			Cost of intervention		
			Must train users		
			Requires physical presence of patient		
			Expense of travel involved		
Pase et al. [31]	Blood biomarkers	Convenient	Taking blood is invasive	III	A
		Inexpensive	Cost of intervention		
			Must train users		
			Requires physical presence of patient		
			Expense of travel involved		
Tahmi et al. [32]	PET biomarkers	Accurate at detecting AD	Cost of intervention	III	A
			Must train users		
			Requires physical presence of patient		
			Expense of travel involved		
Cabinio et al. [33]	Telehealth/VTC/VR	Non-invasive	Cost of intervention	II	A
		Non-stressful	Must train users		
Rajan et al. [34]	Blood biomarkers	Accurate at detecting AD	Taking blood is invasive	III	A
			Cost of intervention		
			Must train users		
			Requires physical presence of patient		
			Expense of travel involved		
Thientunyakit et al. [35]	MRI and PET biomarkers	Non-invasive	Cost of intervention	II	A
		Non-stressful	Requires physical presence of patient		
		Accurate at detecting AD	Expense of travel involved		
Altomare et al. [36]	PET biomarkers	Accurate at detecting AD	Cost of intervention	I	A
		Non-invasive	Requires physical presence of patient		
		Non-stressful	Expense of travel involved		
Desai et al. [37]	PET biomarkers	Accurate at detecting AD	Taking blood is invasive	III	A
			Must train users		
			Cost of intervention		
			Requires physical presence of patient		
			Expense of travel involved		
Lin et al. [38]	Arterial pulse	Non-invasive	Must train users	II	A
		Rapid			
		Inexpensive			
		Accurate at detecting AD			
Liu et al. [39]	Blood biomarkers	Accurate at detecting AD	Taking blood is invasive	II	A
			Must train users		
			Cost of intervention		
			Requires physical presence of patient		
			Expense of travel involved		
Mila-Aloma et al. [40]	PET biomarkers	Accurate at detecting AD	Taking blood is invasive	III	A
			Must train users		
			Cost of intervention		
			Requires physical presence of patient		
			Expense of travel involved		
Sajjad et al. [41]	PET biomarkers	Non-invasive	Cost of intervention	I	A
		Non-stressful	Requires physical presence of patient		
		Repeatable without degradation of results	Expense of travel involved		
Wu et al. [42]	Blood biomarkers	Accurate at detecting AD	Taking blood is invasive	II	A
			Must train users		
			Cost of intervention		
			Requires physical presence of patient		
			Expense of travel involved		
Chun et al. [43]	PET biomarkers	Accurate at detecting AD	Cost of intervention	III	B
			Requires physical presence of patient		
			Expense of travel involved		
Kim et al. [44]	Telehealth/VTC/VR	Accurate at detecting AD	Cost of intervention	II	B
		Inexpensive			

**Table 3 healthcare-11-03013-t003:** Interventions of HIT to diagnose AD.

Intervention Themes and Observations	Frequency
PET biomarkers [23,27,29,32,36,37,40,41,43]	9
Blood biomarkers [17,25,31,34,39,42]	6
Telehealth/VTC/VR [20,24,33,44]	4
AI/machine learning [19,22,28]	3
MRI biomarkers [21]	1
MRI and PET biomarkers [35]	1
Arterial pulse [38]	1
RNA [30]	1
EEG [26]	1
Spectral domain optical coherence tomography [18]	1
	28

Effectiveness indicators of leveraging HIT for the diagnosis of AD.

**Table 4 healthcare-11-03013-t004:** Effectiveness of intervention of HIT to diagnose AD.

Effectiveness Themes and Observations	Frequency
Accurate at detecting AD [17,18,19,20,21,22,23,24,26,28,32,34,35,36,37,38,39,40,42,43,44]	21
Non-invasive [20,21,25,26,27,28,33,35,36,38,41]	11
Non-stressful [25,26,27,28,33,35,36,41]	8
Inexpensive [20,24,26,28,30,31,38,44]	8
Repeatable without degradation of results [26,41]	2
Convenient [31]	1
Rapid [38]	1
Ineffective [29]	1
	53

Barriers to leveraging HIT for the diagnosis of AD.

**Table 5 healthcare-11-03013-t005:** Barriers commensurate with the adoption of HIT to diagnose AD.

Barrier Themes and Observations	Frequency
Cost of intervention [17,18,19,20,21,22,23,27,28,30,31,32,33,34,35,36,37,39,40,42,43,44]	23
Must train users [17,18,19,20,21,22,23,24,26,27,28,30,31,32,33,34,37,38,39,40,42]	21
Requires physical presence of patient [17,18,21,23,27,28,30,31,32,34,35,36,37,39,40,41,42,43]	18
Expense of travel involved [17,18,21,23,27,28,30,31,32,34,35,36,37,39,40,41,42,43]	18
Taking blood is invasive [17,22,30,31,34,37,39,40,42]	9
Not reported [25]	1
Need a procedure for the living [29]	1
	91

## Data Availability

Data from this study can be obtained by contacting the corresponding author.

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
