# Peer review of "Leveraging Technology to Diagnose Alzheimer’s Disease: A Systematic Review and Meta-Analysis"

_healthcare, 2023, doi:10.3390/healthcare11233013_

Round 1

Reviewer 1 Report

Comments and Suggestions for Authors

General Comments

The manuscript offers a comprehensive examination of the interventions available for diagnosing Alzheimer’s Disease (AD). The authors have performed a systematic review and meta-analysis, and their rigorous approach is commendable. Their work holds potential to significantly contribute to the body of literature in this area.

Major Points

1-    The criteria for selecting the 15/28 studies for meta-analysis is not explicit. The reasons for excluding the remaining 13 should be elucidated to enhance the transparency of the methodology. Additionally, at the outset, it was mentioned that in "15/28 studies (54%), sufficient data was provided to conduct a meta-analysis." However, later in the text, there's a reference to a "meta-analysis of similar studies." Could you clarify whether the meta-analysis was conducted using only those 15 studies or if other similar studies were incorporated?

2-    The interventions range from PET scans to telehealth, which vary significantly in terms of methodology, feasibility, and accuracy. It would be beneficial to provide a rationale for including such diverse interventions in a single meta-analysis.

3-    In the section mentioning "The weighted average effect size was d=0.7339, which approaches a large effect size," you claim that the magnitude of the effect is approaching a large effect size. Could you specify which criteria or references you are using to determine the size of this effect? Conventions vary in the literature, so it's crucial for readers to understand your frame of reference.

4-    You mentioned the use of telehealth in diagnosing AD with efficacy comparable to traditional methods. Can you detail the telehealth diagnostic procedures that were reviewed? Furthermore, how do they compare in specificity and sensitivity to traditional face-to-face diagnostic methods?

5-    It's mentioned that interventions were effective in all but one study, yet only 40% of the interventions were noted for their accuracy. How do you reconcile this discrepancy? Can you provide a more detailed analysis of the studies where interventions were deemed ineffective?

6-    The manuscript briefly touches upon AI/machine learning as potential diagnostic interventions. Could you elaborate on the specific methodologies or algorithms that were evaluated? What were the criteria to assess their efficacy and accuracy?

Minor Points

1-      While the paper notes that the weighted average effect size approaches a large effect size, it would be helpful to provide the traditional benchmarks (small, medium, large) for Cohen’s d for context.

2-      Machine learning and Bayesian networking under AI have been briefly mentioned. Given the complexity and variety in these methodologies, more detailed descriptions or specific algorithms used would enhance the paper’s clarity.

3-      In your list of interventions, such as telehealth, AI/machine learning, etc., it would be beneficial to provide clearer definitions or references for each intervention. This will assist readers less familiar with some of these terms to better understand their context and applicability.

4-      In the "Interventions of HIT to Diagnose AD" section, specific percentages and absolute numbers of studies that used various interventions are mentioned. However, in other sections, such as in "Table 4 summarizes the effectiveness indicators observed," only percentages are presented. For better consistency and understanding, it would be beneficial to present both (absolute numbers and percentages) in all relevant sections.

The manuscript is methodically sound, providing a helpful review of the interventions available for AD diagnosis. However, significant concerns need to be addressed to enhance the paper's clarity, validity, and transparency.

I would recommend "accept after minor revision". Once the aforementioned major and minor points are addressed, this paper can be a valuable addition to the journal.

Comments on the Quality of English Language

The manuscript is generally well-written and coherent. However, there are minor grammatical errors and inconsistencies, such as punctuation around percentages. Addressing these will enhance the manuscript's polish and readability. There are no glaring or grave errors that significantly detract from the manuscript's overall quality, but as always, a thorough proofreading is advised.

Author Response

General Comments

The manuscript offers a comprehensive examination of the interventions available for diagnosing Alzheimer’s Disease (AD). The authors have performed a systematic review and meta-analysis, and their rigorous approach is commendable. Their work holds potential to significantly contribute to the body of literature in this area.

Major Points

1-    The criteria for selecting the 15/28 studies for meta-analysis is not explicit. The reasons for excluding the remaining 13 should be elucidated to enhance the transparency of the methodology. Additionally, at the outset, it was mentioned that in "15/28 studies (54%), sufficient data was provided to conduct a meta-analysis." However, later in the text, there's a reference to a "meta-analysis of similar studies." Could you clarify whether the meta-analysis was conducted using only those 15 studies or if other similar studies were incorporated?

** On page 27, we added a sentence of clarification. In 13 studies, effect size was not reported, which eliminated them from the meta analysis.

** The second reference to the 15/28 articles is in the opening sentence of the discussion. A summary of findings is required by PRISMA. The “similar studies” references the 15 articles which reported effect sizes in a manner enabling the meta analysis.

2-    The interventions range from PET scans to telehealth, which vary significantly in terms of methodology, feasibility, and accuracy. It would be beneficial to provide a rationale for including such diverse interventions in a single meta-analysis.

** A meta-analysis like this has not been previously performed. The intent of the article was to show that technologically oriented interventions are available to diagnose AD, and enough of them reported an effect size to calculate an overall effect.

3-    In the section mentioning "The weighted average effect size was d=0.7339, which approaches a large effect size," you claim that the magnitude of the effect is approaching a large effect size. Could you specify which criteria or references you are using to determine the size of this effect? Conventions vary in the literature, so it's crucial for readers to understand your frame of reference.

** Thank you for this comment. We agree that this is important. To help readers understand our calculations, we included the confusion matrix on page 27. This matrix uses a Chi-square, 2x2 format to calculate the overall odds ratio and confidence interval. To show how the effect size was calculated, we included all reported OR, F1, Pearson’s r, and Cohen’s d. A full version of this matrix is listed in Appendix C which includes the weighted average effect size. Calculations were performed by R-studio.

4-    You mentioned the use of telehealth in diagnosing AD with efficacy comparable to traditional methods. Can you detail the telehealth diagnostic procedures that were reviewed? Furthermore, how do they compare in specificity and sensitivity to traditional face-to-face diagnostic methods?

** This is answered on page 27, which included video teleconference, virtual reality, wearable sensors, and telemonitoring.

5-    It's mentioned that interventions were effective in all but one study, yet only 40% of the interventions were noted for their accuracy. How do you reconcile this discrepancy? Can you provide a more detailed analysis of the studies where interventions were deemed ineffective?

** “Effectiveness” was defined on page 4: Cost, efficiency, invasiveness. If a study reported a cost advantage to the intervention, it was deemed “effective” based on the definition. The term “accuracy” was only reported in 40% of the interventions. This does not mean to imply the others were not accurate. It only summarizes what the studies reported.

6-    The manuscript briefly touches upon AI/machine learning as potential diagnostic interventions. Could you elaborate on the specific methodologies or algorithms that were evaluated? What were the criteria to assess their efficacy and accuracy?

** The following AI interventions were reported in Table 1: Bayesian network, machine learning using structural equation modelling (SEM). Prediction performance with SEM was performed using receiver operator characteristic (ROC) analysis to generate an area under the ROC curve and sensitivity and specificity distributions.

Minor Points

1-      While the paper notes that the weighted average effect size approaches a large effect size, it would be helpful to provide the traditional benchmarks (small, medium, large) for Cohen’s d for context.

** We added that the weighted average effect size is medium.

2-      Machine learning and Bayesian networking under AI have been briefly mentioned. Given the complexity and variety in these methodologies, more detailed descriptions or specific algorithms used would enhance the paper’s clarity.

** I disagree. Only three of the 28 studies included AI interventions. These are considerably complex interventions and further elucidation would distract from the overall findings of this study. Information about these interventions is provided in Appendix A.

3-      In your list of interventions, such as telehealth, AI/machine learning, etc., it would be beneficial to provide clearer definitions or references for each intervention. This will assist readers less familiar with some of these terms to better understand their context and applicability.

** We provided some additional information reference AI and machine learning.

4-      In the "Interventions of HIT to Diagnose AD" section, specific percentages and absolute numbers of studies that used various interventions are mentioned. However, in other sections, such as in "Table 4 summarizes the effectiveness indicators observed," only percentages are presented. For better consistency and understanding, it would be beneficial to present both (absolute numbers and percentages) in all relevant sections.

 ** The absolute numbers are reported in the verbiage above Table 4.

The manuscript is methodically sound, providing a helpful review of the interventions available for AD diagnosis. However, significant concerns need to be addressed to enhance the paper's clarity, validity, and transparency.

I would recommend "accept after minor revision". Once the aforementioned major and minor points are addressed, this paper can be a valuable addition to the journal.

Reviewer 2 Report

Comments and Suggestions for Authors

The study is pretty straight forward and accomplishes the goal of summarizing available technologies for the treatment of AD. 

The evaluation regarding the strength and quality of evidence for each intervention is rather vague and might benefit from being defined by more specific criteria. 

Comments on the Quality of English Language

Please go over text for minor spelling errors.

Author Response

Reviewer 2

The study is pretty straight forward and accomplishes the goal of summarizing available technologies for the treatment of AD. 

The evaluation regarding the strength and quality of evidence for each intervention is rather vague and might benefit from being defined by more specific criteria. 

Comments on the Quality of English Language

Please go over text for minor spelling errors.

** A thorough proofreading was performed.

Reviewer 3 Report

Comments and Suggestions for Authors

Author Response

Thank you for sharing your research. The study is of great interest and aligns well with my own research topics. I would like to make some comments and suggestions.

Aspects to be noted

METHODOLOGY

The information contained in the Methodology should be better systematised and the terminology standardised.

** We reviewed the methodology for consistency with the other 43 previously published with this protocol and reported in accordance with PRISMA. We found some items that needed improvement.

In the sub-section Data Item, lines no 123-127 it is stated: “In accordance with the published protocol, we collected the following fields of data at each step: PICOS (participants, intervention, comparison to control, medical outcomes, and study design), observed bias, effect size, sensitivity, specificity, F1, country of origin, statistics used, strength of evidence, quality of evidence, measures of effectiveness (cost, efficiency, invasiveness, etc.), and barriers” For example, in the PICO criteria, the contents of each criterion could have been detailed. Furthermore, about interventions in RESULTS, other terminology is used, such as "experimental intervention" or "intervention topic". Similarly, I believe that the measures of efficiency should be defined first, in the methodology.

** We added more detail in PICOS.

** I did not find any instance of “Intervention topic” and the only place I found “experimental intervention” was in Table 1.

** We better defined the measures of effectiveness.

RESULTS

The systematisation of the tables and appendices, and the selection of table headings makes it difficult to understand the results.

** We made some improvements to the tables’ and appendices’ headings

Tables 3 and 4 should be the first ones. The information between tables 1 and 3 and appendices A and B is repetitive.

** We are sorry, but reporting is dictated by PRISMA. We cannot alter the order of reporting and maintain that we followed the PRISMA guidelines. The information in Tables 1 and 3 and appendices A & B are not repetitive. Table 1 is the PICOS for all studies before a thematic analysis was performed. Table 3 provides the frequency that each intervention theme occurred. Appendix A provides a match to the observation and the theme. Appendix B provides all other observations. There is not a single bit of repeated information other than the list of authors in the first column.

Related to the sample of size, it is 320 ( it should be accompanied by the +/-sd). I think that the sample size should be considered in the first table that refers to the characteristics of the studies selected, not in the appendices.

** We changed the reporting of the sample size, and added sample size in the participant column.

It is difficult to assess what can be extracted from the selected articles because the information is not systematised in the Methodology. In particular, with regard to measures of effects and statistical analysis:

Effect of size sample:

-Lines no 182-183. It is said “Additionally, 26/28 (93%) were quality A (consistent results with sufficient sample sizes, adequate control, definitive conclusions)”. I don't see anything in the Methodology, about what size should be considered sufficient. Further reading of the manuscript is necessary to find more information. In particular, in the line no 208 it is said” The weighted average effect 208 size was ?Ì…=0.7339, which approaches a large effect size”. Where is the "acceptable" limit? Between which values should d range?

** Sufficient sample size depends on the power analysis conducted for each study. If the sample size is below the number identified by the power analysis, then that would not be sufficient.

** Acceptable limits for effect size are predetermined: 0.2=small, 0.5=medium, 0.8=large, 1.2=very large. This was defined by Cohen.

Appendix C. Statistical analysis

Line no 137 (Methodology). “Measures of effect were summarized in tables where they were reported. Odds ratios, correlation coefficients, and the F1 were converted to Cohen’s d. A weighted average effect size, as well as a group specificity, and sensitivity were reported”

**A Cohen’s d was used to report the weighted effect size.

Line no 202 (Results). “The necessary ¿? fields were specificity, sensitivity, F1, Odds Ratio, Pearson’s r, and Cohen’s d for effect size”

** We do not see anything actionable in your observation.

The terms of the heading of appendix C his table does not reflect the results of these parameters outlined in lines 137 or onwards. Nor are the ranges previously known. There are narratives, in this regards, in tables and appendices A and B, but I think a layperson would have difficulty understanding the pathway to a non-invasive and cost-effective technique to confirm a diagnosis of AD. To avoid lengthening the explanation, two similar meta-analysis, which could be used as a model.

1.-Medicine (Baltimore). 2018 May;97(20):e10744. DOI: 10.1097/MD.0000000000010744

2.-The Effect of Non-Stroke Cardiovascular Disease States on Risk for Cognitive Decline and Dementia: A Systematic and Meta-Analytic Review Kayla B. Stefanidis 1 & Christopher D. Askew2 & Kim Greaves 3 & Mathew J. Summers Neuropsychol Rev (2018) 28:1–15 DOI 10.1007/s11065-017-9359-z

**I disagree. We conducted our meta-analysis using plain language. A lay-person would most likely not understand much of the calculations of a meta-analysis. The 10 studies used in the statistical analysis for the meta-analysis reported results in a manner that enabled the calculations.

OTHER QUESTIONS

Line no 63. It is said “Psychological distress has also been attributed to susceptibility to AD and more rapid progression of the disease”. Why is this emphasised and not other risk factors? I do not see this aspect sufficiently developed in the work to suggest the use of another diagnostic methodology that is not face-to-face or invasive for the patient.

**We agree. We removed this sentence from the introduction.

Line no 89. It is said “studies had to use as participant older adults (≤ 50)”. Is it correct?

**We removed the age restriction.

Line no 103. Why was Brain chosen and not other Journals?

**This was our error. Line 103 and the figure have been corrected.

Line no 156 It is said “A kappa statistic was calculated based on the 156 levels of agreement between reviewers (k=0.83, strong agreement). This should be indicated in the methodology.

**We added this to the methodology under the selection process.

Line no 166. The authors specify here, and not in the METHODOLOGY, which ten years have been investigated. In addition, from 2012 (included) to 2022 (included) are 11 years.

** The study was conducted from the middle of 2012 to the middle of 2022. This is 10 years.